# Spinel rGO Wrapped CoV_2_O_4_ Nanocomposite as a Novel Anode Material for Sodium-Ion Batteries

**DOI:** 10.3390/polym12030555

**Published:** 2020-03-03

**Authors:** Rasu Muruganantham, Jeng-Shin Lu, Wei-Ren Liu

**Affiliations:** Department of Chemical Engineering, R&D Center for Membrane Technology, Research Center for Circular Economy, Chung Yuan Christian University, Taoyuan City 32023, Taiwangto42916@gmail.com (J.-S.L.)

**Keywords:** anode, sodium ion batteries, spinel, reduced graphene oxides

## Abstract

Binary mixed transition-based metal oxides have some of the most potential as anode materials for rechargeable advanced battery systems due to their high theoretical capacity and tremendous electrochemical performance. Nonetheless, binary metal oxides still endure low electronic conductivity and huge volume expansion during the charge/discharge processes. In this study, we synthesized a reduced graphene oxide (rGO)-wrapped CoV_2_O_4_ material as the anode for sodium ion batteries. The X-ray diffraction analyses revealed pure-phased CoV_2_O_4_ (CVO) rGO-wrapped CoV_2_O_4_ (CVO/rGO) nanoparticles. The capacity retention of the CVO/rGO composite anode demonstrated 81.6% at the current density of 200 mA/g for more than 1000 cycles, which was better than that of the bare one of only 73.5% retention. The as-synthesized CVO/rGO exhibited remarkable cyclic stability and rate capability. The reaction mechanism of the CoV_2_O_4_ anode with sodium ions was firstly studied in terms of cyclic voltammetry (CV) and ex situ XRD analyses. These results articulated the manner of utilizing the graphene oxide-coated spinel-based novel anode-CoV_2_O_4_ as a potential anode for sodium ion batteries.

## 1. Introduction

Energy storage devices, namely batteries, play a cardinal role in the up-to-date development of portable smart technology to heavy electric energy storage devices [1]. Currently, sodium ion batteries (SIBs) display their unique advantages of abundant Na resources, being economically low cost, and having a similar device strategy as the viable lithium ion batteries [2,3]. However, the developments of appropriate electrode materials are some of the major critical topics for SIBs since the larger ionic radius of Na hinders the electrochemical reaction via graphite as the anode for Na ion storage. Currently, metal-based electrode materials, such as metal oxides [4], metallic selenides [5], bimetallic sulfides [6], and other metal-based electrode materials [7,8,9], have been widely established and developed for high-energy density SIBs applications. Among these, vanadium with mixed metal oxide-based anode materials tends to display improved electrochemical properties because of the higher energy density, existing in a diversity of oxidation states, and offering a considerably wider potential window for Li/Na ion batteries [8,9,10,11]. Nonetheless, vanadate-based anodes for SIBs suffer from rapid capacity decay and poor rate performance as a result of the extreme pulverization and slow kinetics upon the sodiation/desodiation process [11]. These drawbacks have been amended by modified carbonaceous materials.

Graphene is deliberated as a supreme support due to its enormous specific surface area, high mechanical strength, and superior electronic conductivity. Wang et al. [12] described reduced graphene oxide (rGO) as an anode material for SIBs, and it exhibited a reversible capacity of 174.3 mAh g^−1^ at 0.2 C (40 mA g^−1^) over 250 cycles. Wen et al. reported a reversible capacity of 280 mAh g^−1^ at a current density of 20 mA g^−1^ through increasing the interlayer spacing to 4.3 Å [13]. Kumar et al. recently observed that the rGO anode delivered a discharge capacity of 272 mA h g^−1^ at a current density of 50 mA g^−1^ [14]. Thus, the results indicated rGO as a potential electrochemically active material for SIBs. Recently, several researchers have focused on rGO-modified mixed metal oxide-based anode materials for high-rate and long-cycle Na ion storage applications. Wu et al. [15] observed the highest reversible capacity of the NiFe_2_O_4_/rGO(20 wt. %) anode for sodium ion batteries, and the result showed 450 mAh g^−1^ at 50 mA g^−1^ after 50 cycles with good cycling stability. Sekhar et al. [16] reported ZnMn_2_O_4_ with modification of N-doped graphene composite anode materials for Na ion storage. The reversible capacity was found to be 170 mAh g^−1^ at 100 mA g^−1^ over the 1000 cycles. Zhang et al. [17,18] reported the Co_1.8_V_1.2_O_4_/rGO and Mn_2.1_V_0.9_O_4_/rGO materials for Li ion batteries. Li et al. [19] proposed monodispersed graphene nanosheets on a SnO_2_ nanoparticle hybrid electrode material for Li/Na ion batteries. The Na ion storage cell exhibited a high reversible capacity of 314 mAh g^−1^ at 100 mA g^−1^ and showed a stable long-term cycling stability with a capacity retention of 77% after 500 cycles. In our previous work, we reported CoV_2_O_4_ as an anode material for Li ion storage for the first time, and it achieved higher electrochemical performance via carboxymethyl cellulose (CMC) with styrene butadiene rubber (SBR) polymer binder comparable to the conventional PVDF binder [20]. So far, there is no literature available based on CoV_2_O_4_ or rGO-modified CoV_2_O_4_ material for sodium ion battery anodes.

In this work, we demonstrate a facile technique to synthesize rGO-wrapped CoV_2_O_4_ (CVO/rGO) nanocomposite for the first time with a sodium ion storage anode material. The as-prepared materials’ thermal stability, crystal phase structure, and surface morphology are evaluated by TG analysis, XRD, microscopy, and BET analysis. The CVO/rGO sample has high thermal stability, a higher specific surface area, and mesoporous nature. TEM result shows that rGO completely encloses the CVO nanoparticles. The resultant CVO/rGO nanocomposite exhibits higher rate performance and excellent long-term cycling stability. The electrochemical kinetic behavior and storage mechanism are proposed via AC impedance, cyclic voltammetry, and ex situ XRD analyses.

## 2. Experimental Section

### 2.1. Preparation of rGO-Wrapped CoV_2_O_4_ Material

All the regent chemicals were used in the experiment without any further purification of the process. First, 5 wt. % of graphene oxide (GO) was dispersed into 40 mL of methanol (CH_3_OH, 99.9%, TEDIA) solution using an ultrasonication process (amplitude of 50 Hz at 30 min). The obtained solution was noted as Solution A. Afterwards, 0.8731 g of cobalt nitrate hexahydrate (Co(NO_3_)_2_∙6H_2_O, 98%, SHOWA) and 0.7019 g of ammonium vanadate (NH_4_(VO_3_), 99%, Alfa Aesar, 99%) were mixed into Solution A and stirred vigorously for 30 min. Then, an equal ratio of 2.5 mL of nitric acid (HNO_3_, 69.5%, Scharlau) and 2.5 mL of hydrogen peroxide (H_2_O_2_, 35%, SHOWA) was dropwise added into the metal and GO mixed solution and stirred for 1 h. The completely mixed solution was transferred into a 100 mL Teflon-lined stainless steel autoclave and reacted in an oven at 200 °C for 3 days, then naturally cooled to room temperature. The precipitate was washed with ethanol several times and dried at 80 °C. Finally, the as-prepared material was calcined at 500 °C for 8 h in a H_2_/N_2_ atmosphere, and rGO-wrapped CoV_2_O_4_ (CVO/rGO) with a spinel crystal phase structure was obtained. Bare CoV_2_O_4_ (CVO) was synthesized by a similar procedure of the abovesaid steps only without GO in the solution. Figure 1a shows the schematic representation of the rGO-wrapped CoV_2_O_4_ material’s preparation.

### 2.2. Characterizations

The thermal behaviors of the as-prepared materials were investigated using TGA analysis. The ramp rate was maintained at 5 °C/min under an air atmosphere. The crystal phase structure and purity were determined by XRD analysis using a D8 diffractometer (Bruker^®^) with monochromatic CuKα radiation. The operating voltage, current, and wavelength (λ) were 40 kV, 30 mA, and 1.54060 Å, respectively. Diffraction data were recorded in the range (2θ) of 10°–80°. The morphological natures of the as-prepared samples were observed using SEM (Hitachi S-4100) with electron mapping (EDS, X-MAX) and high-resolution TEM (JEM2100) techniques. The specific surface area and porosity were estimated by using A Tristar 3000 accelerated surface area and porosimetry instrument via the N_2_ adsorption/desorption isotherm route. 

### 2.3. Preparation of the Electrode Using Cellulose-Based Polymer Binder

The working electrodes were prepared by mixing 80 wt. % of active material (CVO or CVO/rGO), 10 wt. % of conductive additive (Super P, Timcal^®^), and a 10 wt. % mixture of carboxymethyl cellulose (CMC, 6 wt. %)/styrene butadiene rubber (SBR, 4 wt. %) binder. The CMC/SBR mixture was dissolved in deionized H_2_O. Then, the slurry was uniformly coated on a copper foil. The coated Cu foil was dried at 120 °C for 6 h using an oven. Finally, it was punched into a disc-like electrode with a diameter of 1.2 cm, 1.5–2.0 mg cm^−2^ of active material mass loading, and ~40 mm in thickness. 

### 2.4. Fabrication of the Sodium Ion Cell

The sodium ion storage cell was fabricated using the CR2032 coin-type half cell with sodium metal as the reference electrode and the prepared electrode as the working electrode. Glass fiber (Type A/E, P/N 61630, Pall Corporation) was used as a separator between the working electrode and sodium metal. The electrolyte was composed of 1 M NaClO_4_ (99.99%, Sigma-Aldrich, St. Louis, MO, USA) dissolved in a mixture of ethylene carbonate (EC) and diethyl carbonate (DEC) (3:7 *v*/*v*) with 5 wt. % of fluoroethylene carbonate (FEC). The coin cell assembling procedures were performed using Ar filled glove box (MBraun lab star model) by keeping both the O_2_ and H_2_O levels less than 1 ppm. 

### 2.5. Electrochemical Measurements

The galvanostatic discharge-charge measurements were performed using a constant current and constant voltage programmable AcuTech battery testing system (New Taipei City, Taiwan, Model 750B) in the potential window of 0.01–3.0 V (V vs. Na/Na^+^) under a 20 ± 2 °C temperature at various rates. The cyclic voltammograms (CV) were measured by the electrochemical workstation of the CH Instruments Analyzer (CHI 6273E, Austin, TX, USA) at a scan rate of 0.5 mV/s in the potential rage of 0.001–3.0 V. The AC impedance was carried out in the range of 0.001 Hz–100 kHz at an AC voltage with a 5 mV amplitude.

## 3. Results and Discussion

A schematic view of the simple preparation of the proposed rGO-wrapped CVO (CVO/rGO) is shown in Figure 1a. The as-synthesized CVO/rGO and bare CVO materials crystal phase structure of the XRD patterns are presented in Figure 1b. As exposed in the diffraction peaks (2θ), values of 18.81°, 31.21°, 36.81°, 38.51°, 44.81°, 55.71°, 59.41°, 65.31°, and 72.31° corresponded to the diffraction planes (111), (220), (311), (222), (400), (420), (511), (440), and (553), respectively. All the diffraction peaks were compatible with the standard patterns of face-centered cubic spinel crystal structure with a space group of *Fd*$\overset{¯}{3}$*m* (ZnV_2_O_4_, ICSD# 9009795; Co_2_VO_4_, JCPDS # 73-1633) [20,21,22,23]. The corresponding spinel crystal structure is shown in Figure 1c. 

The thermogravimetric analysis was performed to predict the thermal stability of the resultant products, and the results are shown in Appendix A (Appendix A). The temperature increased from 35 to 800 °C at a heating rate of 5 °C/min in air to monitor the weight loss of the CVO/rGO composite. An increase in the weight with increasing temperature due to the oxidation of the CoV_2_O_4_ material under air could be seen. The weight changes could be classified into the following stages as: (i) the weight loss was about 1%, which occurred between 35–150 °C, and it was speculated that the moisture in the sample was removed; (ii) a 10% of weight increase at a temperature between 150 and 500 °C was observed, which was due to the oxidation reaction of CoO to Co_3_O_4_; and (iii) a final weight increase could be seen at the stage of ~500–650 °C, which was ascribed to the oxidation reaction of V_2_O_3_ to V_2_O_5_. In the range of 350 to 600 °C, the oxidation of carbon occurred [24]. However, the rGO oxidation shielded the further oxidation reaction of V–O matrix compounds during the air atmosphere test. Noticeably, more oxidation with weight increase was observed in the bare CVO sample. Moreover, the CVO/rGO sample showed a very small weight gain after 400 to 800 °C, resulting in the rGO modification promoting the thermal stability of the metal oxide material.

The morphology of the prepared samples was observed using microscopic techniques, namely SEM/TEM analysis. Figure 2a shows the SEM image of the bare CVO material, and it demonstrates the spherical size of the particles with an aggregated particle morphology. Figure 2b illustrates the CVO/rGO sample SEM image. An agglomerated particle morphology could be clearly seen, which was owed to the rGO covered from the CVO nanoparticles. In order to ensure the existence of CVO particles and rGO, we used TEM observation (Figure 2c,d). The prepared CVO particles exhibited two different sizes, namely nano-sized primary irregular particles and ultanano-sized secondary spherical particles (Figure 2c). Both as-prepared samples existed in a similar morphology of particles sizes. The primary particles size was in the range of 20 to 60 nm. Besides, the secondary particle size was found to be within 15 nm. The TEM image of the CVO/rGO sample is shown in Figure 2d, and the inset of Figure 2d displays the low magnification TEM image. Figure 2d demonstrations the fewer layers of graphene sheets obviously wrapping the CoV_2_O_4_ nanoparticles. The primary particles sizes were in the range of 15 to 30 nm and within 10 nm of the secondary particles. The addition of GO solution into the metal oxide mixture solution during the preparation may act as heterogeneous nucleation seeds to facilitate the formation of small crystal grains. Thus, the resultant CVO particles of the CVO/rGO sample showed a smaller size compared to the bare CVO sample. The nano sizes of the CVO particles would buffer the volume changes, and the rGO helped to promote the electrical conductivity, as well as the electrical contact with the active particles during the sodiation/desodiation process.

The as-synthesized CVO material purity and its chemical elements oxidation states were analyzed by the X-ray photoelectron spectroscopy (XPS) and X-ray absorption spectroscopy (XAS) techniques. Figure 3a reveals the survey spectrum of the CVO sample, and the presence of Co, V, and O elements was confirmed. Figure 3b shows the Co 2p core spectra with Gaussian fitting, and it was distinguished into two different sets of binding energies. The major peak binding energies were found to be 781.4 and 798.9 eV and the satellite peaks binding energies around 787.2 and 803.3 eV, which were attributed to the Co 2p3/2 and Co 2p1/2 in 2+ oxidation state in the CVO material. The V2p narrow scan XPS result is shown in Figure 3c. The four deconvolution peaks of V existed in the 3+ and 5+ oxidation states. The V (3+) state peaks were located at 516.33 and 523.63 eV. The very small band peaks were exhibited around 517.05 and 524.88 eV, representing the V 5+ state. The spin-orbit splitting band difference was 7.3 and 7.8 eV for the 3+ and 5+ states. The values were close to earlier reports [17,18,20]. Furthermore, the oxidation state of V was verified by V *K*-edge X-ray absorption near edge structure spectroscopy (XANES) analysis. Appendix A shows the V *K*-edge XANES result of the CVO material with the reference spectra of V, VO_2_, V_2_O_3_, and V_2_O_5_, respectively. The first absorption feature between 5465 and 5475 eV was ascribed to the pre-edge peak owing to the transition of the 1s core level to the 3d states. It was a dipole-prohibited transition, but initiated by the combination of strong 3d–4p mixing and the node of the metal 3d orbitals with the surrounding O 2p orbitals [25]. The lesser intensity of pre-edge peak was ascribed to the more symmetric VO_6_ octahedra in the substituted compounds [26]. The spectra region around 5498.91 eV was similar to the V_2_O_3_ reference peak, which indicated V in the 3+ state. The small band regions were located at 5540.49 and 5567.31 eV, corresponding to V in 5+. Hence, the presence of mixed V 3+ and 5+ was confirmed in the CoV_2_O_4_ materials. Thus, the results of XPS and XAS indicated that the prepared materials exhibited the spin-glass behavior of V in the CVO spinel structure. The functional oxygen groups were analyzed from the core spectra of O 1s, and the result is shown in Figure 3d. The binding energies were located at 530.3 and 531.6 eV, corresponding to the lattice oxygen of the oxygen metal framework (V–O–V) and the hydroxyl group from the moisture absorption on the surface of the sample.

Figure 4a shows the N_2_ adsorption and desorption curves of bare CoV_2_O_4_ and rGO-wrapped CVO nanocomposite materials. The specific surface area was found to be 27 m^2^ g^−1^ for CVO and 35 m^2^ g^−1^ for CVO/rGO samples. The observed surface area was higher than that of some other metal oxide materials [27]. Figure 4b displays the pore volume distribution of the prepared bare CoV_2_O_4_ and CVO/rGO nanocomposite materials. A higher pore volume distribution was observed for the CVO/rGO sample (0.2115 cm^3^/g) compared with bare CVO (0.1569 cm^3^/g), which was presumably CoV_2_O_4_ particles no longer agglomerating into spheres, so the pore volume increased. The pore sizes of the materials were found to be 23.30 and 24.41 nm for bare CVO and CVO/rGO samples, indicating the mesoporous nature. The higher surface area and pore volume promoted the contact area between the material and the electrolyte and provided sufficient space to shield the volume expansion during the electrochemical reaction [28]. 

The electrochemical performances of the as-prepared materials were evaluated as an anode material for SIBs. The first, second, and third sodiation/desodiation profiles of CVO and rGO-wrapped CVO/Na cells are shown in Figure 5a,b at a current density of 100 mA g^−1^, respectively. The discharge/charge platforms became similar in shape; however, the CVO/rGO electrode showed a higher reversible specific capacity. The bare CVO’s reversible discharge capacity was 125 mAh g^−1^, and the initial Coulombic efficiency was found to be 51% (Figure 5a). In addition, the enhancement of reversible discharge capacity was predicted for the CVO/rGO electrode cell. It was found to have a second cycle discharge capacity of 150 mAh g^−1^ and an initial Coulombic efficiency of 48% (Figure 5b). 

Figure 5c displays the cyclic stability test of both prepared samples. Initially, 1–3 cycles were performed at a current density of 100 mA g^−1^, and subsequent cycles were carried out at 200 mA g^−1^. The CVO/rGO electrode cell demonstrated higher capacity with cyclic stability than the bare CVO electrode, which was owed to the improvement of the conductivity of the bare materials. The Coulombic efficiency increased after the initial formation cycles and maintained around 98% over the 1000 cycles of both samples. The initial lower Coulombic efficiency was ascribed to the common issues of metal oxide anode materials due to the decomposition of the electrode active material under the phase conversion reaction during the sodiation process and the formation of the solid-electrolyte interface (SEI) film formation [29,30]. Furthermore, the higher surface area sample (CVO/rGO) hindered the initial Coulombic efficiency more than the bare material because of the side reaction on the electrolyte and electrode. The CVO/rGO cell showed a discharge capacity of 94.66 mAh g^−1^ over the 1000 cycles with a retention of 81.6% at 200 mA g^−1^. The bare CVO electrode cell exhibited a discharge capacity of 69.41 mAh g^−1^ with a retention of 73.5 % over the 1000 cycles at 200 mA g^−1^. The observed reversible capacity of the CVO/rGO material illustrated higher performance than those of earlier reports of other oxide-based anode materials for SIBs [31,32]. The following reasons could be given for the remarkable cyclic stability of the prepared electrode materials: (i) the ultrasmall CVO nanoparticles shortened the Na^+^ ion diffusion distance; (ii) the appreciable surface area and porous nature with the combination of conductivity-based polymer binder possessed an electrochemically active thin stable layer of SEI, which may have improved the structural stability by acting as a buffer layer for the volume changes during the Na ion insertion and extraction processes [33,34,35]; (iii) the quantity of the total number of atoms that were close or on the surface was augmented when the particle size was reduced [36]; consequently, the accessible electroactive surface area was increased, resulting in improved electrochemical reaction; and (iv) the rGO additionally promoted the excellent electrical conductivity and the flexible higher surface area. Thus, the results showed this to have higher electrochemical stability and capacity as anode materials for SIB’s. The high-rate Na ion storage capability test was performed at different current densities of 0.1, 0.2, 0.4, 0.8, 1.6, and 3.2 A g^−1^, and the result is shown in Figure 5d. The prepared materials were fine candidates for high-rate sodium ion batteries. The CVO/rGO electrode delivered average specific capacities of 149, 127, 120, 99, 77, and 54 mAh g^−1^ at current densities of 0.1, 0.2, 0.4, 0.8, 1.6, and 3.2 A g^−1^, respectively. When the current density returned to 0.1 A g^−1^, the electrode exhibited an average capacity of 114 mAh g^−1^, revealing an appreciable high-rate capability.

The electrochemical kinetic behavior of the electrode cell was evaluated by the electrochemical impedance spectra (EIS) technique after five cycles at 100 mA g^−1^. Both as-prepared electrode cells’ Nyquist plots derived from EIS are shown in Figure 6a. The data were fitted by an equivalent circuit (see Figure 6a, inset), and the resultant fitting parameters are exposed in Table 1. The corresponding charge transfer resistance R_ct_ values of CVO/rGO and bare CVO were 17 and 86 Ω. The abundantly lower resistance of the CVO/rGO cell extremely promoted the transfer of electrons and Na ions during sodiation/desodiation reactions [17,18,20]. The diffusion coefficient of sodium ions can be calculated from the following equation:D = (R^2^*T*^2^)/(2*A*^2^*n*^4^F^4^*C*^2^σ^2^)(1)
where R is the gas constant, *T* is the absolute temperature, *A* represents the electrode area, n is the number of electrons during charge/discharge, F is the Faraday constant, *C* is the concentration of sodium ions, and σ is the Warburg factor. The Warburg factor (σ = Z’/ω^−1/2^) can be explained by the diffusion barrier layer’s impedance and semi-infinite diffusion impedance in the lower frequency slope after the semicircle. The σ was inversely proportional to the diffusion coefficient of Na^+^ [10]. Hence, the R_ct_ and σ were ascribed to the kinetic reaction of the cell. Figure 6b shows the CVO and CVO/rGO cells’ linear fitting relationship plot between Z_re_ and the reciprocal square root of the lower angular frequencies. The slope values of the CVO and CVO/rGO electrodes were 5.81 and 2.55, respectively. The diffusion coefficients of Na ions could be estimated according to the above formula, which were 2.77 × 10^−11^ and 1.44 × 10^−10^ cm^2^ s^−1^, respectively. The CVO/rGO cell demonstrated higher diffusivity than that of bare CVO, which meant it boosted the conductivity of the electrode. Thus, the CVO/rGO anode demonstrated finer capacity and rate capability. Besides, this material’s electrochemical performance was lower than that of Li ion battery anodes [20]. To understand the reason, the electrochemical reaction mechanism was described via CV and ex situ XRD analysis. 

Figure 7a exhibits CV curves of the CVO electrode cell for the initial three cycles at a scan rate of 0.5 mV s^−1^. The cathodic peaks at 1.1 V and 0.4–0.7 V were attributed to the Na ion insertion into CoV_2_O_4_ and conversion reaction that occurred to form CoO, as well as further reduction of CoO to Co [20,37]. The absence of the reduction band around 0.4 V after the first cycle represented the irreversible formation of SEI film. The broad band of the anodic peak was observed around 1.0 to 2.1 V, which was ascribed to Na ion extraction from the V–O matrix and the oxidation reaction from Co to CoO. The first cycle’s cathodic current was reduced, and the second and third cycles’ peaks were different than the first cycle. This represented the irreversible reaction of the first sodiation process. For the second and third cycles, the cathodic peak of 1.1 V was shifted to a lower voltage range, and reduction of CoO shifted to a higher voltage range, which was owed to the structural rearrangement in the initial charge process [38]. This showed that the CV curves at the second and third cycles almost overlapped, signifying the excellent electrochemical stability of the material. 

Figure 7b displays different cut-off voltage states of the galvanostatic discharge/charge curves for the analysis of the ex situ XRD test at the first cycle, and Figure 7c shows the corresponding ex situ XRD patterns of the CVO sample at different charge/discharge states in the first cycle. The nature of the spinel structure was observed in the fresh electrode. The electrodes were adapted to the sodiation/desodiation process at various cut-off voltage ranges. Remarkably, the diffraction peaks of the CVO electrodes showed similar XRD patterns compared to that of fresh electrode, which suggested the incomplete electrochemical reaction. The Co standard X-ray diffraction peaks (ICSD # 44989, space group of Fm-3m) of 2θ values were 44.37° (111), 51.59° (200), and 75.92° (220), respectively. The standard pattern of CoO (ICSD # 29082, space group: F-43m) exhibited 34.17° (111), 39.66° (200), 57.24° (220), 68.39° (311), and 71.84° (222), respectively. Obviously, the diffraction peaks of Co were similar to fresh electrodes, and they were difficult to distinguish. However, the diffraction peaks of the fresh electrode slightly shifted to a lower angle during the sodiation/desodiation process (Appendix A). At a discharge of 0.01 V, very small characteristic peaks appeared at the position of 44.5°. It was thought that the partial conversion reaction of CVO was observed during the first sodiation cycle. This phenomenon implied the confirmation of the insertion/extraction of Na^+^ during the discharge/charge process. Hence, the electrochemical performance showed lower capacities compared to the conversion reaction of Li ion storage [20]. In addition, the deficient electrochemical performance might have been the Na_2_O or 2NaVO_2_ formation during the sodiation/desodiation process [39]. The Na_2_O or NaVO_2_(O3) phases acted as an ionic, as well as electronic insulator and prevented further conversion reaction into CVO material [39,40]. Overall, the as-synthesized rGO-wrapped CVO material showed a remarkable enhancement of the electrochemical performance, and the following factors could be described: (i) the rGO with complete coverage of CoV_2_O_4_ nanoparticles provided adequate electrode-electrolyte contact areas for more Na^+^ ion transfer across the interface and shortened the length of Na^+^ ion diffusion; (ii) the rGO aided efficiently accommodating the volume change during the sodiation/desodiation process; and (iii) the rGO-modified CVO improved the conductivity.

## 4. Conclusions

We successfully synthesized the rGO-wrapped CoV_2_O_4_ nanocomposite by the solvothermal technique. The cubic spinel crystal structure was revealed. The CVO/rGO sample had a specific surface area of 35 m^2^/g, and it was comparably higher than that of bare CVO (27 m^2^/g). The as-synthesized materials were both applied to sodium ion storage anodes for the first time. The results showed excellent long-term cyclic stability and good rate performance. The rGO-wrapped CoV_2_O_4_ electrode cell exhibited a higher reversible discharge capacity of 150 mAh g^−1^ and retained 81.6% of capacity after 1000 cycles at 200 mA g^−1^. For the rate performance, it demonstrated a capacity of 54 mAh g^−1^ at 3200 mA g^−1^. Moreover, the bare CVO electrode cell obtained 73.5 % of retention after 1000 cycles at 200 mA g^−1^. The discharge capacity was found to be 43 mAh g^−1^ at the current density of 3200 mA g^−1^. Normally, the Na ion insertion/extraction process in metal oxide-based anodes causes large volume changes, leading to the pulverization process and a loss of electrical contact. Consequently, in this work, the rGO sheets wrapped with CVO nanoparticles served as a highly conductive framework to sustain the electrical contact from the CVO nanoparticles to the current collectors and improved the capacity and cycling stability due to their high electrical conductivity and large surface area. This work opens a prospect to develop this potential candidate for vanadium-based metal oxide anode materials for high performance SIB applications.

## Figures and Tables

**Figure 1 polymers-12-00555-f001:**
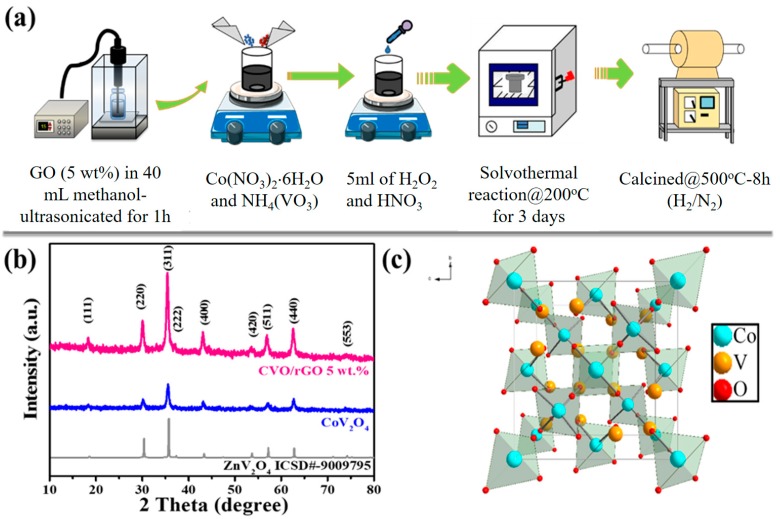
(**a**) Schematic illustration of the synthesis route for reduced graphene oxide (rGO)-wrapped CoV_2_O_4_ material; (**b**) XRD patterns of bare CoV_2_O_4_ and rGO-wrapped CoV_2_O_4_ materials; and (**c**) schematic crystal structure of spinel CoV_2_O_4_.

**Figure 2 polymers-12-00555-f002:**
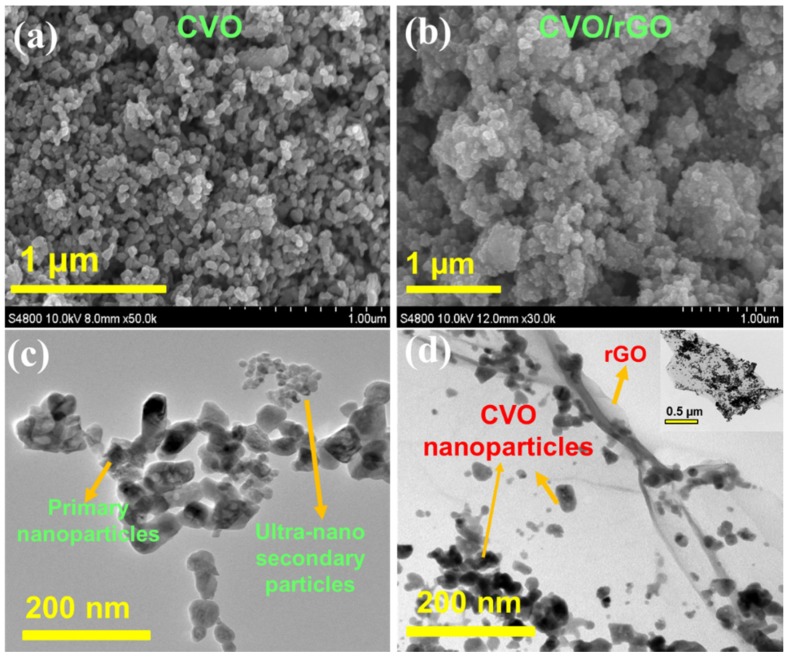
(**a**,**b**) SEM images and (**c**,**d**) TEM images of bare CoV_2_O_4_ and rGO-wrapped CoV_2_O_4_ materials (inset of Figure 2d is a large magnification TEM image of CVO/rGO sample).

**Figure 3 polymers-12-00555-f003:**
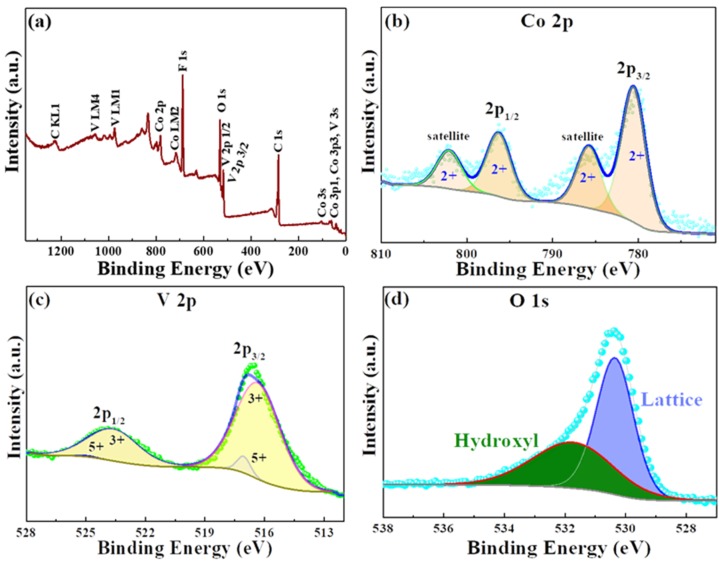
XPS spectra of the CoV_2_O_4_ sample: (**a**) wide range spectra; (**b**) Co 2p core spectra; (**c**) V2p core spectra; and (**d**) O1s core spectra.

**Figure 4 polymers-12-00555-f004:**
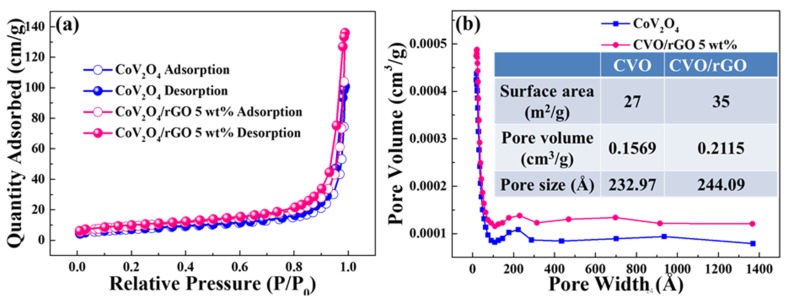
(**a**) Nitrogen adsorption isotherms and (**b**) the BJH desorption pore volume distributions of bare CoV_2_O_4_ and rGO-wrapped CoV_2_O_4_ materials; inset: the table shows the estimated values.

**Figure 5 polymers-12-00555-f005:**
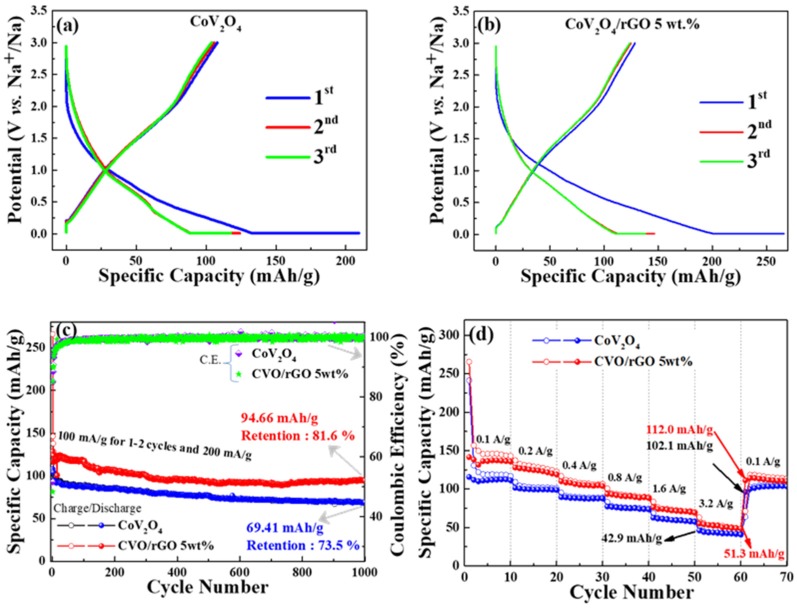
Electrochemical performance of Na ion storage: (**a**,**b**) discharge/charge profile of bare CoV_2_O_4_ and rGO-wrapped CoV_2_O_4_ electrodes; (**c**,**d**) cyclic stability and rate capability of bare CoV_2_O_4_ and rGO-wrapped CoV_2_O_4_ electrodes.

**Figure 6 polymers-12-00555-f006:**
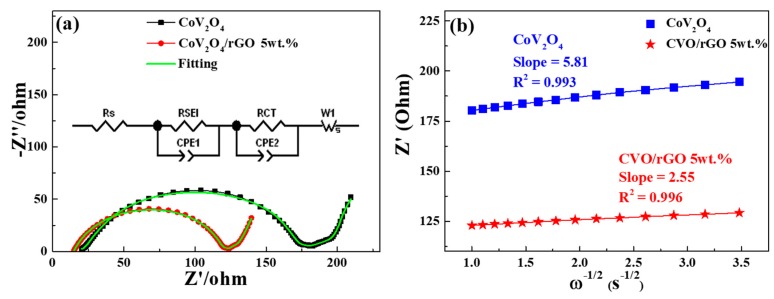
(**a**) Electrochemical impedance spectra (Nyquist plots); inset: the equivalent circuit used to fit the EIS; and (**b**) the liner fit of the relationship between Z′ and ω^−1/2^ at low frequencies of the bare CoV_2_O_4_ and rGO-wrapped CoV_2_O_4_ electrodes.

**Figure 7 polymers-12-00555-f007:**
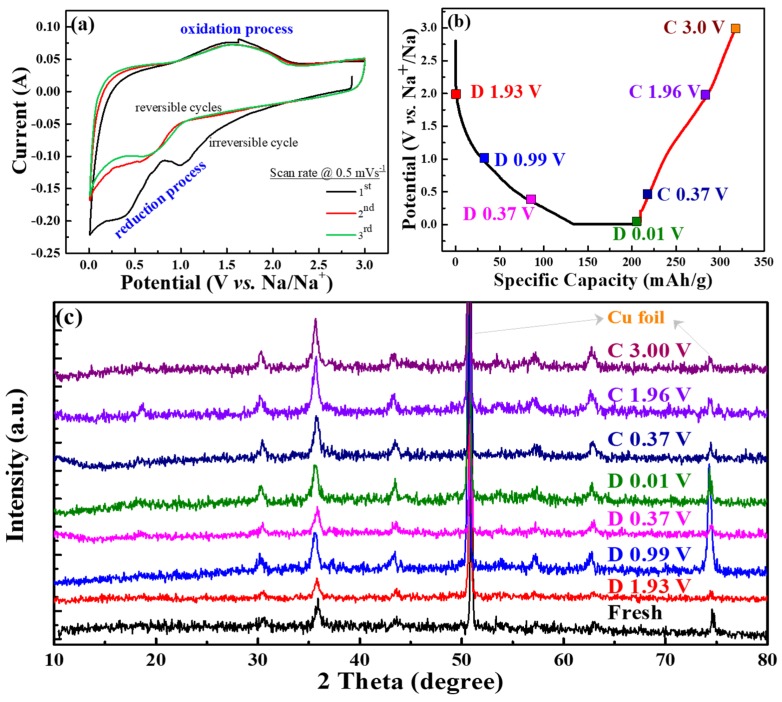
Sodiation/desodiation mechanism analysis of the CoV_2_O_4_ material electrode cell: (**a**) cyclic voltammetry (CV) curves in the potential window range of 0.01–3.0 V at a scan rate of 0.5 mV s^−1^; (**b**) discharge/charge profile at various voltage states for ex situ studies; and (**c**) ex situ XRD analysis.

**Table 1 polymers-12-00555-t001:** Impedance parameters and diffusion coefficients of the equivalent circuit.

Parameters	CoV_2_O_4_	CVO/rGO 5 wt. %
R_S_ (Ω)	10.54	14.32
R_SEI_ (Ω)	137.20	102.80
CPE1-T	1.2746 × 10^−5^	2.0206 × 10^−5^
CPE1-P	0.86534	0.83426
CPE2-T	0.015158	0.06819
CPE2-P	0.13263	0.24671
W1-R	96.72	162.3
W1-T	50.87	149.8
W1-P	0.67235	0.7404
R_CT_(Ω)	86.20	17.10
Slope	5.81	2.55
R^2^	0.993	0.996
Diffusion Coefficient (cm^2^/s)	2.77 × 10^−11^	1.44 × 10^−10^

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
