# Peer review of "Spinel rGO Wrapped CoV2O4 Nanocomposite as a Novel Anode Material for Sodium-Ion Batteries"

_polymers, 2020, doi:10.3390/polym12030555_

Round 1

Reviewer 1 Report

In this paper, reduced graphene oxide wrapped CoV2O4 material was demonstrated as promising anode for long-term cycling in sodium-ion batteries. The work is overall quite novel (first demonstration) and the results are well presented. Some minor edits/revisions are suggested prior to publication:

  1. In Fig 1b, the words (311) and (222) are slanted. They should be vertical.
  2. In Fig 3b, both of the doublet peaks are labeled 2+. I believe that's a mistake, please check.
  3. In Fig 5c, the right axis label Coulombic efficiency is cut off.
  4. Some relevant review papers on sodium batteries should be referenced as well: Energy Environ. Sci. 11, 2673-2695 (2018), Chem Rev 119,5416-5460 (2019).
  5. How can the cycling performance be further improved? Some discussion can be added.

Author Response

Comment 1 of reviewer 1:

In Fig 1b, the words (311) and (222) are slanted. They should be vertical.

Response of Comment 1 of reviewer 1:

We thank to the reviewer suggestion. We have corrected the mistake in the revised Fig. 1.

Comment 2 of reviewer 1:

In Fig 3b, both of the doublet peaks are labeled 2+. I believe that's a mistake, please check.

Response of Comment 2 of reviewer 1:

In Fig.3b, the binding energies peaks around 787.2 and 803.3 eV are represented the satellite peaks of Co 2p3/2 and Co 2p1/2 in 2+ oxidation state of Co. It is not a mistake. Thanks to the reviewer’s clarification of the comment.

Comment 3 of reviewer 1:

In Fig 5c, the right axis label Coulombic efficiency is cut off.

Response of Comment 3 of reviewer 1:

We apology for this mistake. In the revised version, we have corrected the mistake in the revised Fig. 5(c).

Comment 4 of reviewer 1:

Some relevant review papers on sodium batteries should be referenced as well: Energy Environ. Sci. 11, 2673-2695 (2018), Chem Rev 119,5416-5460 (2019).

Response of Comment 4 of reviewer 1:

Thank you for reviewer’s suggestion. In the revised version manuscript, we have included the following references with the reference numbers.

[34] Lee, B.; Paek, E.; Mitlin, D.; Lee, S. W. Sodium Metal Anodes: Emerging Solutions to Dendrite Growth, Chem. Rev. 2019, 119, 5416.

[35] Zhao, Y.; Adair, K.; Sun, X. Recent developments and insights into the understanding of Na metal anodes for Na-metal batteries, Energy Environ. Sci., 2018, 11, 2673.

Comment 5 of reviewer 1:

How can the cycling performance be further improved? Some discussion can be added.

Response of Comment 5 of reviewer 1:

We extremely thank to the reviewer for the constructive comments. The following description about the discussion of how to enhance cyclic performance have been added in the revised manuscript.

“The following reasons can be described for the remarkable cyclic stability of the prepared electrode materials. There are (i) the ultrasmall CVO nanoparticles provide shortens the Na+ ion diffusion distance, (ii) the appreciable surface area and porosity nature with combination of conductivity-based polymer binder possess the electrochemically active thin stable layer of SEI, which may improve the structural stability by interim as a buffer layer for the volume changes during Na-ion insertion and extraction processes [33-35], (iii) the quantity of the total number of atoms that lie close or on the surface is augmented when the particle size is reduced [36]. Consequently, the accessible electroactive surface area is increased, resulting in improved electrochemical reaction, and (iv) the rGO additionally promotes with excellent electrical conductivity and the flexible higher surface area. Thus, the results showed the higher electrochemical stability and capacity as anode materials for SIB’s.”

We hope reviewer satisfy with our response.

Reviewer 2 Report

The authors present a structural and electrochemical study of a nanocomposite material formed from reduced graphene oxide wrapping cobalt- and vanadium-containing spinel nanoparticles. They compare the results to a similarly-formed material that does not have the graphene wrapping. The results show that the inclusions of the graphene wrapping improves the performance of the material for sodium ion batteries. The implications of this study are interesting, but the results are not as well presented as they should be, and the poor English grammar impedes understanding. This work is suitable for publication in Polymers after the comments below are considered.

MAJOR COMMENTS

  1. The XPS results described in Lines 164 and following are just for the CVO samples, not the graphene wrapped samples. Similarly, the CV and ex-situ XRD results in Lines 255 and following are only for the CVO samples. In both of these cases, comparison of the two types of materials would strengthen the conclusions of the manuscript.
  2. The entire manuscript should be read and edited by a native speaker of English before final publication. The grammatical issues are not just a matter of clarity in the manuscript, but actually impede the understanding in a number of places.
  3. It would be useful to include the fit lines on the Nyquist plot in Figure 6(a). Additionally, the fit results in Table 1 are incomplete. What are the values of fit parameters for CPE1, CPE2, and W1? What are the uncertainties in these fit parameters? The diffusion coefficient should in principle be able to be found directly from these fit parameters.

MINOR COMMENTS

  1. The vertical axis on Figure S1 is not labeled clearly. Numbers above 100% represent weight gain, not loss. The axis should be labeled “Weight Change (%)” instead. Additionally, the results would be clearer if graphed as lines rather than the overlapping symbols.
  2. In Line 157, the words “wrapped with” should be replaced by “wrapping”.
  3. The numerical adsorption/desorption results in Lines 190 and following are reported with more significant figures than I suspect are warranted. What is a reasonable estimate of the uncertainty in these values? Even if the uncertainties are not reported, the numerical results should be rounded to the number of significant figures that are reasonable to report.
  4. The discussion of coulombic efficiency in Lines 209-215 are about Figure 5(c), not 5(a-b). So it would be more clear to move this discussion after the discussion of cyclic stability in Lines 216-224.
  5. The right-hand side vertical axis labels (for coulombic efficiency) in Figure 5(c) are cut off.
  6. The horizontal axis label of Figure 6(b) should be “ω-1/2 (s-1/2)”.

Author Response

Reviewer: 2

The authors present a structural and electrochemical study of a nanocomposite material formed from reduced graphene oxide wrapping cobalt- and vanadium-containing spinel nanoparticles. They compare the results to a similarly-formed material that does not have the graphene wrapping. The results show that the inclusions of the graphene wrapping improves the performance of the material for sodium ion batteries. The implications of this study are interesting, but the results are not as well presented as they should be, and the poor English grammar impedes understanding. This work is suitable for publication in Polymers after the comments below are considered.

Comment 1 of reviewer 2:

The XPS results described in Lines 164 and following are just for the CVO samples, not the graphene wrapped samples. Similarly, the CV and ex-situ XRD results in Lines 255 and following are only for the CVO samples. In both of these cases, comparison of the two types of materials would strengthen the conclusions of the manuscript.

Response of Comment 1 of reviewer 2:

We thank to the reviewer for the precious suggestion. Our intense is to identify this material anode behavior of sodium-ion batteries for the first-time. Thus, we have performed the bare material CV and ex-situ XRD analysis. For the improvement purpose, we used to modify the rGO into the CVO material and we checked the storage performances. The role of rGO is to act boosting the conductivity (confirmed via EIS) and to buffer the volume changes of metal oxide during the electrochemical reaction as results to exposed the higher electrochemical performance of rGO-wrapped CVO sample. We do believe rGO will not affect the electrochemical reaction mechanism between Na ions and CoV2O4 anode. It might be not necessary to carry out XPS and ex-situ XRD measurements of CVO/rGO in our study.

We hope reviewer satisfy with our response.

Comment 2 of reviewer 2:

The entire manuscript should be read and edited by a native speaker of English before final publication. The grammatical issues are not just a matter of clarity in the manuscript, but actually impede the understanding in a number of places.

Response of Comment 2 of reviewer 2:

Thank you to the reviewer suggestion. In revised manuscript, we have carefully corrected the grammatical errors and typo-errors by native English speakers.

Comment 3 of reviewer 2:

It would be useful to include the fit lines on the Nyquist plot in Figure 6(a). Additionally, the fit results in Table 1 are incomplete. What are the values of fit parameters for CPE1, CPE2, and W1? What are the uncertainties in these fit parameters? The diffusion coefficient should in principle be able to be found directly from these fit parameters.

Response of Comment 3 of reviewer 2:

We thank to the reviewer for valuable comment. The following EIS spectra and the parameters of CPE1, CPE2 and W values have been included in the revised manuscript.

Comment 4 of reviewer 2:

The vertical axis on Figure S1 is not labeled clearly. Numbers above 100% represent weight gain, not loss. The axis should be labeled “Weight Change (%)” instead. Additionally, the results would be clearer if graphed as lines rather than the overlapping symbols.

Response of Comment 4 of reviewer 2:

Thank you for reviewer’s comments. We agreed with reviewer’s comment. We have changed and the revised following Fig.S1 added in the revised manuscript.

Comment 4 of reviewer 2:

In Line 157, the words “wrapped with” should be replaced by “wrapping”.

Response of Comment 4 of reviewer 2:

Thank you for reviewer’s suggestion. In revised version, we have changed the word “wrapped” to “wrapping”.

Comment 4 of reviewer 2:

The numerical adsorption/desorption results in Lines 190 and following are reported with more significant figures than I suspect are warranted. What is a reasonable estimate of the uncertainty in these values? Even if the uncertainties are not reported, the numerical results should be rounded to the number of significant figures that are reasonable to report.

Response of Comment 4 of reviewer 2:

Thank you for reviewer’s suggestion. In revised version, we have rounded the specific surface area values such as 26.94 changed to 27 and 34.67 changed to 35.

Comment 4 of reviewer 2:

The discussion of coulombic efficiency in Lines 209-215 are about Figure 5(c), not 5(a-b). So it would be more clear to move this discussion after the discussion of cyclic stability in Lines 216-224.

Response of Comment 4 of reviewer 2:

Thank you for reviewer’s comments. In revised version, we have changed the line “209-215” to “213-220”.

Comment 4 of reviewer 2:

The right-hand side vertical axis labels (for coulombic efficiency) in Figure 5(c) are cut off.

Response of Comment 4 of reviewer 2:

We apology for this mistake. In the revised version, we have corrected the mistake in the revised Fig. 5(c).

Comment 4 of reviewer 2:

The horizontal axis label of Figure 6(b) should be “ω-1/2 (s-1/2)”.

Response of Comment 4 of reviewer 2:

We apologies for this mistake. We have corrected the mistake in the revised manuscript.
